# Enhancement of Rotator Cuff Healing with Farnesol-Impregnated Gellan Gum/Hyaluronic Acid Hydrogel Membranes in a Rabbit Model

**DOI:** 10.3390/pharmaceutics13070944

**Published:** 2021-06-24

**Authors:** Yen-Hung Lin, Sheng-I Lee, Feng-Huei Lin, Guan-Xuan Wu, Chun-Shien Wu, Shyh-Ming Kuo

**Affiliations:** 1Institute of Biotechnology and Chemical Engineering, I-Shou University, Kaohsiung City 82445, Taiwan; glen0212@gmail.com; 2Department of Pharmacy and Master Program, Tajen University, Yanpu Township 90741, Taiwan; 3Department of Orthopedics, Pao-Chien Hospital, Pingyung 90064, Taiwan; 4Department of Biomedical Engineering, I-Shou University, Kaohsiung 82445, Taiwan; isu1040055@cloud.isu.edu.tw (S.-I.L.); isu10901002D@cloud.isu.edu.tw (G.-X.W.); 5Department of Biomedical Engineering, National Taiwan University, Taipei 10617, Taiwan; double@ntu.edu.tw; 6Center for General Education, I-Shou University, Kaohsiung 82445, Taiwan; cswu@isu.edu.tw

**Keywords:** rotator cuff tendon tears, farnesol, collagen synthesis, hyaluronan, hydrogel membranes

## Abstract

Most rotator cuff (RC) tears occur at the bone–tendon interface and cause disability and pain. Farnesol, a sesquiterpene compound, can exert antioxidative and anti-inflammatory effects and promote collagen synthesis. In this rabbit model, either commercial SurgiWrap membrane or hydrogel membranes containing various compositions of gellan gum, hyaluronic acid, and farnesol (hereafter GHF membranes) were applied to the tear site, and the repair of the cuff was examined 2 and 3 weeks afterward. The designed membranes swelled rapidly and adsorbed onto the tear site more readily and closely than the SurgiWrap membrane. The membranes degraded slowly and functioned as both a barrier and a vehicle of slow farnesol release during the repair period. Farnesol enhanced collagen production in myoblasts and tenocytes, and interleukin 6 and tumor necrosis factor α levels were modulated. Gross observations and histological examinations indicated that the GHF membranes impregnated with 4 mM farnesol resulted in superior RC repair. In sum, the slow release of farnesol from hydrogel membranes can be beneficial in the repair of RC injuries.

## 1. Introduction

Shoulder pain is the most common functional problem involving the upper limbs, and rotator cuff (RC) tears are a common cause of shoulder dysfunction. The typical clinical symptoms of these injuries are pain or poor mobility caused by the inflammation of the synovium. Often, surgical repair is required if symptoms persist after conservative medication or rehabilitation [1]. Knowledge regarding the pathology and healing pattern of the RC, use of biological repair techniques and better suture anchors, and gradual rehabilitation after surgical repair of RC tears can lead to satisfactory outcomes [2]. However, regardless of the procedures used, healing failure after surgical repair remains a critical problem and complication. Numerous studies have examined whether biological agents such as platelet-rich plasma, transforming growth factor-β1 (TGF-β1), fibroblast growth factor-2, and stem cells can augment healing after RC repair [3,4]. TGF-β1 plays a crucial role in the healing process and promotes collagen production, angiogenesis, and extracellular matrix (ECM) formation. However, the use of growth factors raises various concerns with regard to optimal amount, time points, and half-life, as does the sequential emergence of factors involved in the healing process. Therefore, the long-term delivery of growth factors through controlled release from biomaterial vehicles can enhance RC repair [5,6,7].

The enthesis, the connective tissue located at the junction of tendons and bones, consists of four areas composed of distinct cell types and matrices: tendons, fibrocartilage, mineralized fibrocartilage, and bones [6]. The unique structure of the enthesis minimizes muscle–bone stress attributable to gradual changes in the alignment and composition of tissues. Interventions should be developed to overcome the complexity and low healing potential of the enthesis in RC tears [8].

Farnesol, an organic 15-carbon sesquiterpene alcohol produced by *Candida albicans*, possesses anti-inflammatory, antimicrobial, and tumor-related apoptosis-inducing characteristics [9]. Ku et al. demonstrated in vivo that farnesol downregulated essential inflammatory cytokines, such as interleukin (IL)-1β, IL-6, and tumor necrosis factor α (TNF-α), as well as the TNF-α/IL-10 ratio [10]. Our previous study indicated that farnesol improved wound healing by reducing oxidative stress and inflammation and enhancing collagen production [11,12]. Taken together, these findings suggest that farnesol can modulate connective tissue and ECM synthesis and promote wound healing in tissue engineering applications.

Gellan gum (GA), a linear, anionic polysaccharide secreted by *Pseudomonas elodea*, functions as a stabilizer, thickening agent, and gelling agent in various foods. Recently, GA has been investigated for its use as a drug delivery agent, cell carrier, guided bone regeneration material, and wound dressing material in biomedical engineering applications because of its biocompatibility and low cytotoxicity [5,13].

Hyaluronic acid (HA), a key glycosaminoglycan constituent of the ECM, consists of the repeated disaccharides of D-glucuronic acid and N-acetyl-D-glucosamine. Numerous clinical studies have indicated that the subacromial injection of HA can alleviate inflammation and prevent adhesion at the RC tear site. During the RC repair process, many inflammatory cells infiltrate and form fibrous scar tissue, which impairs tendon–bone healing. Because of its anti-inflammatory effect, HA has been used to promote tendon–bone healing in ligament reconstruction [14]. Moreover, its effects are mediated and regulated through its binding interactions with specific cell-associated receptors such as CD44. CD44 is involved in HA-modulated cell adhesion, migration, and proliferation and can enhance tissue growth and repair. In addition, HA promotes the expression of type I collagen as well as enhancing the viability and proliferation of tendon-derived cells [15,16].

To the best of our knowledge, no study has applied farnesol, an organic anti-inflammatory agent that promotes collagen production, to RC tear repair. Therefore, the present study investigated whether the application of farnesol-impregnated hydrogel membranes containing HA or GA (hereafter GHF membranes) can enhance RC tear repair in a rabbit model, comparing its efficacy with that of SurgiWrap membrane, a type of bioresorbable sheet approved by the US Food and Drug Administration.

## 2. Materials and Methods

Farnesol (CAS No: 4602-84-0), HA (MW: 9 × 10^5^ Da, CAS No: 9004-61-9), gellan gum (GA: 500 kDa, CAS No: 71010-52-1), 3-4,5-dimethylthiazol-2-yl-2,5-diphenyltetrazolium bromide (MTT), and dimethyl sulfoxide (DMSO) were purchased from Sigma (St. Louis, MO, USA). Dulbecco’s modified Eagle’s medium, fetal bovine serum, trypsin, streptomycin, and penicillin were obtained from Gibco. Commercially available SurgiWrap membrane (MAST Biosurgery, San Diego, CA, USA) used as an anti-adhesion barrier in gastroenterology was applied to the comparison group. All chemicals used in this study were of reagent grade.

Animal experiments conducted in this study were approved by the Institutional Animal Care and Use Committee of I-Shou University, Taiwan (Approval No: IACUC-ISU-108-035, approval date: 12 February 2020).

### 2.1. Fabrication of GHF Hydrogel Membranes

Three kinds of membranes containing 30% GA and 40% HA (weight ratio) were prepared without farnesol, with 2 mM farnesol, and with 4 mM farnesol, respectively, and their basic properties were evaluated. The weight ratio of GA and HA was determined according to our previous studies [17]. GA and HA were dissolved in 40 mL of distilled water to yield a transplant solution. Subsequently, 2 or 4 mM farnesol was added and stirred for 1 h at 40 rpm. The solution was then poured into a glass dish and evaporated in an oven at 37 °C for 24 h to obtain a dry membrane. This dry membrane was further crosslinked with 15 mM 1-ethyl-3(3-dimethylaminopropyl)-carbodiimide/N-hydroxysuccinimide (EDC/NHS) for 8 h at room temperature. The crosslinked membrane was washed with 95% ethanol three times to remove any residual EDC/NHS and then dried at room temperature. The surface microstructure of the membranes was examined using scanning electron microscopy (SEM). The basic properties of these membranes were determined by repeating the experiments three times, and the means and standard deviations were calculated.

### 2.2. Characterization of GHF Membranes

#### 2.2.1. Water Content and Swelling Ratio Measurement

The water content (WC) and swelling ratio (SR) of the GHF membranes were determined by immersing the membranes in phosphate-buffered saline (PBS) with a pH of 7.4 at room temperature. At predetermined time intervals, the wet membranes were blotted using filter paper to remove water adhered to the membrane surface. The WC and SR of the membrane were determined as follows:WC (%) = (W_W_ − W_d_)/W_d_ × 100%
SR (%) = (W_W_ − W_d_)/W_W_ × 100%
where W_w_ and W_d_ are the weights of the wet and dry membranes, respectively.

#### 2.2.2. Mechanical Property Measurement

These prepared hydrogel membranes swelled considerably in the aqueous solution and thus, could not be mounted firmly onto the load cell used in measurements; therefore, we conducted the mechanical strength test under dry conditions. The membranes were cut into pieces (1 cm × 6 cm), and the tensile strength of the membranes was measured up to the rupture point of the membranes. The mechanical parameters of the membranes were obtained using a material testing system (MTS; Eden Prairie, MN, USA) at a crosshead speed of 5 mm/min.

#### 2.2.3. In Vitro Degradation Test

The in vitro degradation of the prepared membranes was examined by incubating the membranes in 10 mL of PBS (pH 7.4) in a vial that was then placed on a shaker set at 40 rpm and 37 °C. At predetermined times, the membrane was removed from the incubation medium, washed with distilled water, dried, and weighed, after which another 10 mL of fresh PBS was added to the vial and the degradation test was continued. The degradation profiles were obtained as the cumulative weight loss of the membranes.

#### 2.2.4. In Vitro Release Test

The in vitro release of farnesol from the GHF membranes was assessed. A circular GHF membrane (3.5 cm diameter) was placed in a 50 mL tube containing 5 mL of PBS. The tube was then placed on a shaker with the temperature and shaking rate set at 37 °C and 40 rpm, respectively. At designated time points, 1 mL of PBS was collected and centrifuged at 12,000 rpm for 10 min and 1 mL of fresh PBS was added into the tube for continuous shaking. The amount of farnesol present in the supernatant was determined through high-performance liquid chromatography (HPLC; Aglient-1100). The mobile phase of HPLC analysis consisted of 80% acetonitrile and 20% distilled water, and the flow rate was set as 20 μL/min. The in vitro release rate was calculated as follows:In vitro release (%) = [(total amount of farnesol − residue of farnesol)/total amount of farnesol] × 100%

### 2.3. In Vitro Cell Viability

The effects of farnesol at various concentrations on the viability of C2C12 myoblasts and tenocytes were determined using the MTT assay. The cells (5 × 10^3^ cells/mL) were seeded in 96-well plates for 24 h and incubated with different concentrations of farnesol. Next, 20 μL of the MTT solution (5 mg/mL) was added to each well, and the cells were incubated for an additional 3 h. The formazan precipitate formed was dissolved in 200 μL of DMSO, and the solution was vigorously mixed to dissolve the dye. Absorbance of the sample in each well was measured at 570 nm by using a multiplate reader. The control group was 0 mM of farnesol.

Cell viability was also assessed through a live/dead cell assay (Invitrogen, Carlsbad, CA, USA). In brief, 1 mL of PBS solution containing 4 μM ethidium homodimer-1 (EthD-1) assay solution (2.5 μL/mL) and 2 μM calcein acetoxymethyl solution (1 μL/mL) was prepared. This assay solution (100 μL) was added to the culture, and the mixture was placed in 5% CO_2_ for 15 min at 37 °C. The sample was examined under a fluorescence microscope (Olympus IX71, Japan) at excitation filters of 494 nm (green, calcein) and 528 nm (red, EthD-1).

### 2.4. Quantification of Collagen Production

C2C12 myoblasts or tenocytes (5 × 10^4^ cells/mL) were seeded in a 6-well plate for 24 h. The cells were then treated with or without various concentrations of farnesol for 24 or 48 h. Quantification of collagen was performed according to the manufacturer’s instructions (Sircol Soluble Collagen Assay Kit, Biocolor, Carrickfergus, Northern Ireland). Absorbance was measured at 555 nm by using a multiplate reader.

### 2.5. Anti-Inflammatory Test with H_2_O_2_-Induced Cells

The cells (5 × 10^4^ cells/well) were seeded in a 6-well plate for 24 h and preincubated with 1 mM H_2_O_2_ for 1.5 h. The cells were then treated with various concentrations of farnesol (0–0.6 mM). After 24 h incubation, the medium was collected and centrifuged at 2000× *g* for 10 min to obtain cell-free supernatant, which was then used to perform the enzyme-linked immunosorbent assay (ELISA) for IL-6 and TNF-α. The levels of IL-6 and TNF-α were quantified using the Elabscience Mouse IL-6 ELISA Kit (Minneapolis, MN, USA) and the Elabscience Mouse TNF-α ELISA kit (USA), and absorbance was measured at 450 nm according to the manufacturer’s protocol. The anti-inflammatory effects of the membranes on IL-6 and TNF-α were evaluated by comparing the IL-6/TNF- α ratio under each treatment.

### 2.6. In Vivo Experiments for the Assessment of RC Tear Repair

#### 2.6.1. Animals and Experimental Design

RC tear was induced in twelve 15-week-old male New Zealand rabbits. Surgical repair was performed by a single surgeon. The rabbits were randomized into the control group (tears not covered with membranes, Group A; *n* = 3) and the experimental groups: Group B (tears covered with SurgiWrap membrane; *n* = 3), Group C (tears covered with the GHF membrane impregnated with 2 mM farnesol; *n* = 3), and Group D (tears covered with the GHF membrane impregnated with 4 mM farnesol; *n* = 3). Postoperation, all the rabbits were given free access to food and water.

In vivo animal experiments were performed over 3 weeks (Figure 1B). The RC of the rabbit forelimb was cut to induce a RC tear. The lesion site was then sutured and covered with membrane. In weeks 2 and 3, six rabbits from each group were randomly selected and sacrificed. Their RC tear samples were harvested for histopathological analysis, and the contents of type I and type III collagen were assayed through both a collagen assay and polarized microscopy examination. Induction and repair of RC tears were performed under anesthesia with xylazine (5–10 mg/kg, intramuscular injection). During the procedures, clinical signs of pain, salivation, and abnormal behavior were carefully monitored.

#### 2.6.2. Histopathological Analysis

In weeks 2 and 3 after the RC tear repair, the RCs were harvested and the RC tissues were fixed in 10% neutral buffered formalin. Next, the RC samples were dehydrated in graded ethanol solutions, cleared in xylene, embedded in paraffin blocks, and cut into 5 μm thick sections. Masson’s trichrome staining was performed for histopathological examination and the assessment of collagen changes from membrane repair. ImageJ software (Version 1.50; National Institutes of Health, Bethesda, MD, USA) was used to measure the collagen content in each group. The software color settings remained constant throughout the analysis of connective tissue (stained in blue) in each sample. Samples were evaluated at 40× magnification, and calculations were repeated in four microscopic fields.

#### 2.6.3. Picrosirius Red Stain with Polarization Microscopy

In addition to Mason’s trichrome staining, numerous alternative methods are available for visualizing collagen in histopathological sections. In the present study, we performed picrosirius red staining with standard polarization microscopy to determine the content of type I and type III collagen in the RC tear site. In brief, hematoxylin and eosin-stained sections were covered with picrosirius red solution (PSR, Abcam, Cambridge, UK) for 60 min at 40 °C. The slide was then rinsed with acidified water, dehydrated, and cover slipped by using a toluene-based mounting medium. The PSR-stained slide was examined under an Olympus microscope (IX-71) with linear polarizers. The presence of type I and type III collagen was indicated by strong yellow-red birefringence and weak greenish birefringence, respectively. The lamp intensity was kept constant during image capture, and ImageJ software was used to quantify the collagen content in each group. Samples were evaluated at 40× magnification, and calculations were repeated in three microscopic fields.

#### 2.6.4. Histological Assessments

We also modified a scale system to evaluate the histological repair of RC tears [18]. Table 1 presents the five criteria, namely collagen fiber density, collagen fiber orientation, tendon–bone interface, vascularity, and ratio of type I to III collagen, which were semi-quantitatively graded. The total score was the sum of scores of these five variables, with a higher score indicating better healing results. Scoring was performed by an orthopedic surgeon who was blinded to treatment details (three independent images, *n* = 3).

### 2.7. Mechanical Test of RC

To measure the breaking strength of the repaired RC tear 6 weeks postoperation, the RC was dissected. Parts of the scapula, humerus, joint capsule, and RC were retained for convenience of fixation with clamps (*n* = 3). The mechanical strength of the RC was measured up to the point of the rupture of the joint capsule and RC. The mechanical parameters of the RC were calculated automatically by using an MTS (MTS; Eden Prairie, MN, USA) at a crosshead speed of 5 mm/min.

### 2.8. Statistical Analysis

All values are expressed as means ± standard errors of the mean. Significant differences between the experimental and control groups were examined using one-way analysis of variance. All analyses were performed using IBM SPSS Statistics for Windows, version 22 (IBM Corp., Armonk, NY, USA). A *p* value of <0.05 was considered statistically significant.

## 3. Results

### 3.1. Basic Properties of the GHF Membranes

The GHF membranes were expected to rapidly reach a steady hydration state, have moderate mechanical strength, and degrade slowly in the release of the impregnated farnesol, thereby promoting the healing of RC tears. As shown in Figure 2A,B, the water content and swelling ratio of the GHF and SurgiWrap membranes reached their hydration equilibrium within 60 min. The water content and swelling ratio of the GHF membranes were significantly higher than those of the SurgiWrap membrane. Notably, the GHF membrane impregnated with 4 mM farnesol had the highest water content and swelling ratio and quickly reached steady equilibrium (within 10 min). The GHF membranes were malleable, allowing smooth affixation to the surgical site, indicating their ease of use in clinical settings, including surgical procedures (Figure 3A). By contrast, the SurgiWrap membrane, which had a lower water content and swelling ratio, could not effectively cover the surgical site.

When membranes (e.g., SurgiWrap) are used in surgical procedures, they typically must be maintained for approximately 1 week to ensure the antiadhesive effect. In the present study, the hydrogel membranes were used as vehicles for the slow release of farnesol to enhance RC tear healing in a rabbit model. As shown in Figure 2C, the GHF membranes degraded slowly and lost 20–40% of their initial weight after they had been placed on a shaker for 10 days. The membranes impregnated with 4 mM farnesol degraded rapidly and completely after they were placed on a shaker for 14 days. This rapid degradation may be ascribable to the water-insoluble component of farnesol and the inhibition of the crosslinking reaction of the EDC/NHS agent. In addition, the SEM images indicated that the surface morphology differed with varying concentrations of farnesol (Figure 3B). However, the SurgiWrap membrane degraded more slowly than did the GHF membranes; this can be explained by its low water content and swelling ratio. Notably, the hydrophobic nature of the SurgiWrap membrane increased the difficulty of use in the surgical procedures. As shown in Figure 2D, the GHF membrane impregnated with 4 mM farnesol rapidly released about 60% of farnesol in 24 h incubation under shaking at 40 rpm, after which it slowly released farnesol until day 9. Figure 3B shows the SEM images of the GHF membranes before (upper lane) and after they were immersed in PBS solution for 24 h (lower lane). After immersion in PBS solution, the integrity of the SurgiWrap membrane remained favorable, and longitudinal surface roughness was noted. This membrane consisted of 70:30 poly(L-lactide-co-D, L-lactide). The polylactide composition strengthened the membrane, keeping the structure intact after the 25-day shaking test at 40 rpm. Furthermore, the hydrophobic characteristic of the polylactide material slowed down the membrane degradation, reduced the water content, and increased the difficulty of its manipulation during operation (Figure 3A). The image of the GHF membranes before immersion in PBS shows an irregular surface on which some of the farnesol is observable. This impregnated farnesol on the surface was released, yielding a smoother surface morphology after 24 h immersion in PBS solution (Figure 3B). In addition, the hydrophilic nature of the HA increased the water content and the manageability of the GHF membranes during the operation, although they degraded more rapidly than did the SurgiWrap membrane. The Young’s modulus of the GHF and SurgiWrap membranes in a dry state ranged from 29.6 to 40.7 MPa. Relative to the SurgiWrap membrane, the GHF membrane impregnated with 4 mM farnesol had significantly higher mechanical strength but less elongation (Table 2).

### 3.2. Assessment of Cell Viability through the MTT Assay and Live/Dead Cell Staining

The effects of farnesol on the viability of C2C12 myoblasts and tenocytes are shown in Figure 4. No significant reduction in the viability of these cells was detected after 24 h incubation in <0.4 mM farnesol. The half maximal inhibitory concentrations of farnesol on C2C12 myoblasts and tenocytes were approximately 0.57 and 0.5 mM, respectively. Regarding the viability of cells treated with farnesol, cell morphology and a live/dead cell assay revealed that the cells gradually shrunk, as indicated by a shortening of the pseudopodia, and detached from the surface under treatment with higher concentrations of farnesol. More dead cells were observed at a higher concentration of farnesol. When the concentration was greater than 0.6 mM, farnesol exerted stronger cytotoxic effects on C2C12 myoblasts and tenocytes; this result is consistent with those from the MTT assay.

### 3.3. Effects of Farnesol on Collagen Production In Vitro

This study examined the ability of farnesol to promote collagen production and exert anti-inflammatory effects. Collagen production was quantified using the Sircol Soluble Collagen Assay. The C2C12 myoblasts did not exhibit a notable increase in collagen production during the initial 24 h treatment. After treatment for 48 h, 0.2–0.8 mM farnesol significantly increased collagen production (approximately 11.1 to 37.9 times). Similarly, a higher concentration of farnesol triggered greater collagen secretion in tenocytes; however, the tenocytes secreted considerably less collagen than did the C2C12 myoblasts. In short, farnesol promoted collagen production in these cells (Figure 5A,B).

### 3.4. Anti-Inflammatory Effects of Farnesol

Our previous study showed in vivo that farnesol reduced IL-6 levels in models of sunburned skin and third-degree burns [18]. Skeletal muscle is an essential source of IL-6 [19]. The IL-6 level in myoblasts was significantly lower in the H_2_O_2_-induced groups treated with 0.2–0.6 mM farnesol; this finding is consistent with those from our previous study on skin [12,20,21]. In addition, this result indicates that farnesol can reduce IL-6 levels both in vitro (in myoblasts) and in vivo, and that this reduction may be attributable to its anti-inflammatory effects. Although mechanisms through which IL-6 secretion is modulated during muscle regeneration have yet to be delineated, inflammatory cytokines such as TNF-α and IL-1 are believed to be involved [22]. IL-6 has been established to be associated with neuroinflammation and plays a key role in the development of pathological pain [23]. Adverse effects of IL-6, including atrophy promotion and muscle wasting, have been reported. Moreover, IL-6 signaling has been correlated with the stimulation of hypertrophic muscle growth and myogenesis [24]. The present results indicate that farnesol reduced H_2_O_2_-induced IL-6 secretion in C2C12 myoblasts (Figure 5C). Therefore, to maintain the effects of IL-6 in skeletal muscle and myoblasts, the appropriate application of farnesol to the skeletal muscle and tendons is crucial. Farnesol can be used after the onset of severe or persistent muscle/tendon injury and pain, such as muscle rupture, muscle strain, and myotendinitis. It can also be used to treat chronic pain and inflammation in muscles or tendons.

In the present study, farnesol at concentrations of 0.2–0.4 mM significantly reduced TNF-α levels in myoblasts. TNF-α has been reported to be an upstream modulator of IL-6 production in septic conditions [25] and cultured skeletal muscle cells, although mechanisms through which TNF-α promotes muscle regeneration remain unclear [26]. Our finding suggests that TNF-α can indirectly stimulate muscle regeneration by increasing IL-6-related signaling [22]. However, TNF-α has been shown to exert complex effects during muscle injury or contusion [27]. In addition to the association with IL-6 secretion in myoblasts, TNF-α has been found to upregulate IL-1β secretion and indirectly facilitate the recruitment of immune cells [28]. IL-6 has been demonstrated to downregulate TNF-α secretion in skeletal muscle [29,30]; this could be one of the reasons treatment with 0.6 mM farnesol caused a slight reversal in increase compared with treatment with 0.4 mM farnesol (Figure 5D). During exercise, a large amount of IL-6 is released from the skeletal muscle into circulation. The corresponding amount of TNF-α released is relatively low, implying that IL-6 plays a more crucial role in systemic responses to muscle activity and injury [31]. Similarly, 0.2 and 0.4 mM farnesol also exerted anti-inflammatory effects on tenocytes by significantly reducing the expression of IL-6 and TNF-α (Figure 5E,F). In sum, the results on the levels of proinflammatory cytokines in C2C12 fibroblasts and tenocytes suggest that a farnesol concentration of 0.4 mM may be appropriate in vitro for reducing the levels of both IL-6 and TNF-α, and farnesol could be used to treat acute muscle injury or persistent chronic muscle pain.

### 3.5. Gross Inspection of RC Tear Healing

Figure 6 shows the gross views of RC tears after 2 and 3 weeks of healing. The membranes impregnated with farnesol (2 and 4 mM) were more easily manipulated and attached onto the RC tear site during the coverage procedure. In general, the phases of muscle healing include inflammation, regeneration, and remodeling. During the inflammatory phase, macrophages infiltrate the injured muscle and participate in muscle regeneration [32]. Muscle regeneration usually peaks 2 weeks after injury and begins declining 3 to 4 weeks after injury. Crucial events during muscle healing include connective tissue remodeling and revascularization (the first sign of muscle regeneration is essential to its success). As shown in Figure 5A, little regenerated tissue around the site of the RC tears without membrane coverage were observed. Moreover, a smaller amount of regenerated tissue was noted on the site of the tears covered with the SurgiWrap membrane (Figure 6B). The group in which RC tears were covered with the farnesol-impregnated membranes exhibited higher amounts of regenerated connective tissue around the tear site than did the blank and SurgiWrap groups after 2 weeks of healing. Some fibrous tissue was noted in these groups (Figure 6C,D). After 3 weeks of healing, more regenerated connective tissue covered the RC tear site in the farnesol-treated groups, indicating a higher degree of healing (Figure 6G,H). Moreover, newly generated vessels were observed at the sites of the tears that were covered with membranes impregnated with 4 mM farnesol (Figure 6H). By contrast, the tear sites covered with SurgiWrap exhibited redness and less regenerated connective tissues (Figure 6F). This may be ascribable to polylactide degradation in the membrane.

### 3.6. In Vivo Histological Assessments

In most cases, the initial stage of RC tear injuries is characterized by acute inflammation, with the accumulation of hemorrhage and leukocytes and the synthesis of bioactive factors. This triggers the proliferation of tenocytes, the migration of tenocytes to the wound, and the increased synthesis of collagen and vascular structures to form immature fibrovascular tissue. In general, the duration of this stage is between 4 and 7 days, and the proliferation or repair phase begins thereafter [33]. At this point, the synthesis of type I and type III collagen is the defining event in the repair process. Studies conducted on tendons have shown that the increased formation of type III collagen continues for weeks to months after injury [34]. However, the more stable type I collagen begins to dominate and participate in the remodeling stage, which is characterized by a reduction in fibrovascular tissue cells and better organization of collagen fiber structures with linear orientation and crimping. If the RC tear site contains random, less oriented tendon fibers and fiber cross-links, scar tissue is formed. This tissue can result in reduced elasticity and durability, increasing the possibility of injury recurrence [35].

Histological findings of the RC tears of the various groups are shown in Figure 7. We mainly focused on the enthesis and evaluated the repair of the RC after injury with or without hydrogel membrane coverage. The repair site of the RC tears without membrane coverage (blanks) exhibited a loose fiber arrangement at the tendon rupture site. Few regenerated collagen fibers and little vascularization were observed, and tendon–bone interdigitation did not appear after 2 weeks of healing (Figure 7A). Similar to those in the blank group, the RC tears covered with the SurgiWrap membrane exhibited loose fibrous tissue, less vascularization, irregular arrangement of collagen fibers at the enthesis, and poor alignment of collagen fibers (Figure 7B). However, the RC tears covered with the membranes impregnated with 2 mM farnesol demonstrated increased vascularization and a slight increase in type I collagen synthesis. Although loose fiber alignment was noted, the alignment of collagen fibers was better than that in the blank and SurgiWrap groups. The inflammation observed is attributable to the rapid degradation of the hydrogel membrane (Figure 7C). The RC tears covered with the membranes impregnated with 4 mM farnesol showed better healing, as indicated by greater collagen synthesis and better collagen fiber alignment, as well as more favorable tendon–bone interdigitation at the end of the RC tear (Figure 7D). Revascularization was observed over this 2-week healing period; however, the rapid degradation of the hydrogel membrane led to the continuation of the inflammatory response. After the 3-week healing period, a greater number of collagen fibers was noted around the tear site in the blank group (Figure 7E). In addition, increased vascularization was observed. The results correspond to the normal repair process at this postinjury stage. At the same period, healing in the RC tears covered with SurgiWrap was comparable to that in the blank group. However, less vascularization was observed; this may be ascribable to the blocking effect of SurgiWrap, the primary purpose of which is to prevent adhesions and the slowed regeneration of blood vessels. The tear site was still not fully interdigitated to the bone; a moderately sized space around the tear site remained (Figure 7F).

Compared with those in the blank and SurgiWrap groups, the RC tears covered with the membrane impregnated with 2 mM farnesol exhibited a greater number of regenerated vessels, as well as greater collagen synthesis around the tear site. This was conducive to tenocyte proliferation and healing after injury. Collagen production by tenocytes facilitated remodeling in the tear site, yielding a better aligned collagen fiber structure (Figure 7G). The RC tears covered with the membrane impregnated with 4 mM farnesol exhibited the most favorable healing (Figure 7H). In line with the gross observation (Figure 6H), considerably increased revascularization was noted around the tear site. In addition, increased collagen production was observed and the collagen fibers were gradually aligned, indicating that remodeling was ongoing but not yet complete.

### 3.7. Picrosirius Red Staining for Collagen Assessment

Picrosirius red staining is a crucial staining method used to examine collagen networks in various tissues. Under polarized light, collagen bundles appear green or red and can be easily differentiated, thereby allowing quantitative morphometric analysis. In the present study, strong yellow-red and weak greenish birefringence was assigned to type I and type III collagen, respectively. Uninjured tendon almost entirely consists of type I collagen. When the tendon is injured, a greater proportion of type III collagen is observed in the early phase of healing [36]. This type III collagen is gradually replaced by type I collagen as tissue remodeling and healing commence. Thus, we evaluated changes in collagen composition in the RC tear sites to compare the reparative effect of the hydrogel membranes. As shown in Figure 8A, the blank group exhibited few newly produced type I or type III collagen fibers. Moreover, fiber continuity was not yet established, as indicated by a moderately sized space around the tear site after 2 weeks of healing. The tear sites covered with the SurgiWrap membrane showed slightly better healing, as indicated by a dense arrangement of type I collagen at the tear site; however, poor interdigitation was still observed at the enthesis. The presence of type III collagen around the RC indicated the ongoing healing process. However, the collagen fibers were irregularly distributed (Figure 8B). As shown in Figure 8C, the tears covered with the membrane impregnated with 2 mM farnesol exhibited an increase in loosely structured regenerated type III collagen. The abundance of regenerated type III collagen indicated enhanced healing around the tear site after a 2-week period. The tears covered with the membrane impregnated with 4 mM farnesol had an organized structure; the density of type I collagen fibers at the end of the RC tear was higher than that in any other experimental group (Figure 8D). At this point—2 weeks postinjury—the tear was still not fully healed, and less interdigitation was observed. After the 3-week healing period, the RC tear sites covered with SurgiWrap was replete with type III collagen fibers (Figure 8E) that were loose and irregularly aligned, as is typical in the normal healing process. In addition, less type I collagen was observed, indicating that a longer period of time is required for type III collagen to be replaced by type I collagen. The RC tears covered with the farnesol-impregnated membranes exhibited greater synthesis of collagen (mainly type III and type I) and increased bridging of collagen fibers at the interface (Figure 8G,H). Notably, in the tear sites covered with the membranes impregnated with 4 mM farnesol, thick, elongated, longitudinally oriented collagen fibers were visible, indicating satisfactory repair. In general, an increase in type III collagen and collagen changes and turnover indicate progress in healing, including in cases of tendon injuries [36]. As mentioned, type III collagen was present in the initial stage of healing and gradually transformed into type I collagen, indicating that healing was complete.

Figure 9 shows the mechanical testing of the repaired RC tear covered with the hydrogel membranes 6 weeks postoperation. The RC tears covered with the membrane impregnated with 4 mM farnesol had a similar range of breaking force with the RC tears covered with SurgiWrap, this force was comparable to normal breaking force, demonstrating good healing of the RC tear. By contrast, the membrane impregnated with 2 mM farnesol exhibited healing results inferior to those in the blank group. RC tears typically heal in approximately 6–12 weeks. Thus, the mechanical strength test may have been performed too early in the healing process (week 6) considering that full-thickness RC tears were induced. Nevertheless, we can conclude that the membranes impregnated with 4 mM farnesol had the highest strength force, indicating superior healing outcomes.

In this study, we also used a modified grading system to evaluate RC tear repair based on histological observations. Specifically, collagen fiber density and orientation, tendon–bone interface interdigitation, vascularity, and the ratio of type I to type III collagen were examined. As shown in Table 3 higher scores were obtained at the 3-week period, indicating healing progress. At the 2-week healing period, the RC tears covered with the membranes impregnated with 2 and 4 mM farnesol exhibited higher scores: 8.33 and 10.67 points, respectively. Inflammation was observed and collagen synthesis began, indicating that the repair process was faster in these two groups. However, the collagen fibers were not well oriented. After 3 weeks of healing, these two groups still had higher scores (14.33 and 18.67 points, respectively) than did the blank and SurgiWrap groups. The tear site covered with the membrane impregnated with 2 mM farnesol exhibited some vascular regeneration, which promoted collagen synthesis. Moreover, the distribution of revascularization or angiogenesis near the tear sites covered with the membranes impregnated with 4 mM farnesol corresponded more closely to the remodeling stage. The tear site covered with the farnesol-impregnated membranes showed increased collagen synthesis, and the alignment of the collagen fibers became more regular over time. The overall proportion of type I collagen increased, and the collagen fibers became densely arranged, changes that indicated the completion of the healing process. Taken together, the results from the gross examination and the histological observations indicated superior healing in tears covered with membranes impregnated with 4 mM farnesol.

To determine whether the GHF hydrogel membranes could be used as a farnesol carrier to treat full-thickness RC tears, we compared their efficacy with that of the SurgiWrap membrane. We selected SurgiWrap over the more commonly used Seprafilm because the high HA content (nearly 66 wt%) of Seprafilm causes the membrane to radially lose its integrity and strength during hydration. The rapid and substantial degradation of Seprafilm in aqueous solutions might not allow tendons to heal within a reasonable period of time. Thus, we elected to examine the SurgiWrap membrane, which has a longer degradation period. The hydrophobic nature of SurgiWrap, which consists of a polylactide material, reduced the objectivity of its comparison with the GHF membranes to some degree. Overall, healing in the GHF-membrane-covered tears was superior to that in the uncovered and SurgiWrap-covered groups.

## 4. Conclusions

In this study, hydrogel membranes impregnated with farnesol, an anti-inflammatory sesquiterpene compound that can promote collagen production, were applied to RC tears. Farnesol exerted anti-inflammatory effects and promoted collagen production in both C2C12 myoblasts and tenocytes. The GHF membranes swelled rapidly, facilitating their manageability during the surgical procedures. Furthermore, the slow release of farnesol facilitated healing through the enhanced production and transformation of collagen fibers, as well as through orientation changes in the collagen fibers. In addition, inflammation was alleviated through the regulation of proinflammatory cytokines. The findings indicate that GHF membranes can potentially be used as a vehicle of farnesol release to accelerate RC tear repair. However, their suitability for such applications should be verified through additional in vivo studies.

## Figures and Tables

**Figure 1 pharmaceutics-13-00944-f001:**
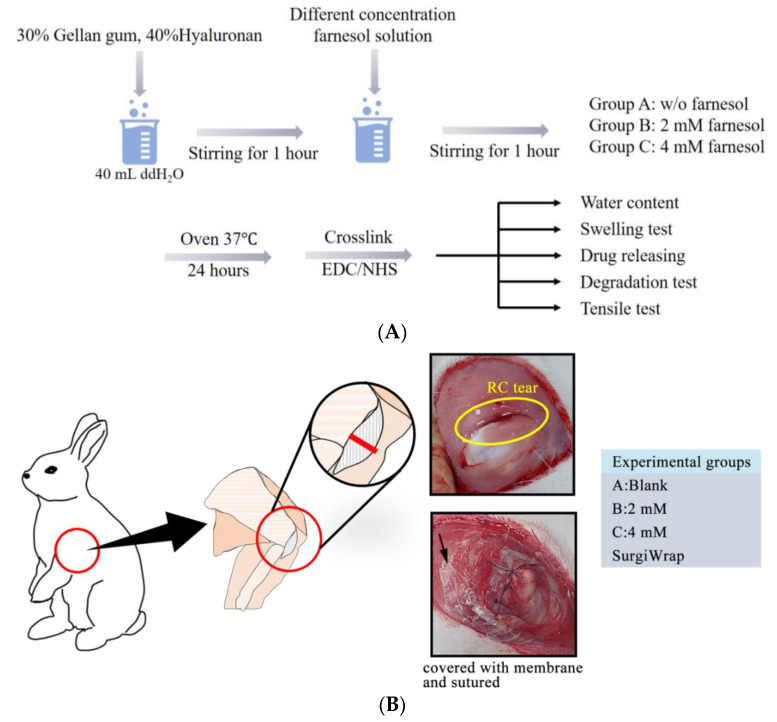
(**A**) Schematic of GHF hydrogel membrane preparation and characterization; (**B**) experimental designs of in vivo studies. A full-thickness RC tear was inflicted on the front limb of each rabbit and then closed using a 3-0 Prolene suture. Subsequently, the tear was either covered with or without the hydrogel membrane. The comparison group comprised rabbits whose RC tears were covered with a SurgiWrap membrane.

**Figure 2 pharmaceutics-13-00944-f002:**
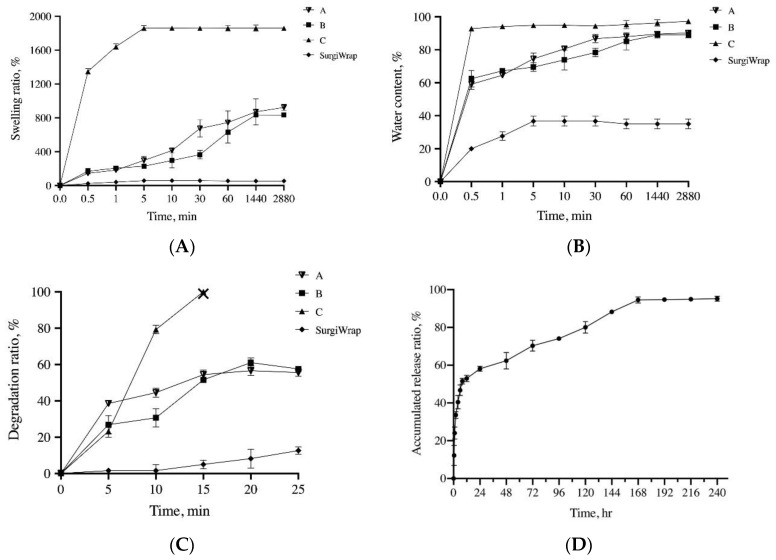
Basic properties of the GHF membranes; (**A**) water content; (**B**) swelling ratio, and (**C**) the degradation profiles of GHF and SurgiWrap membranes in PBS solution with a pH of 7.4 over 24 h; (**D**) the profile of the release of farnesol from the GHF membrane impregnated with 4 mM farnesol in PBS solution with a pH of 7.4 over a 10-day period. (Means ± S.D) A: membrane impregnated 0 mM farnesol, B: membrane impregnated 2 mM farnesol, and C: membrane impregnated 4 mM farnesol.

**Figure 3 pharmaceutics-13-00944-f003:**
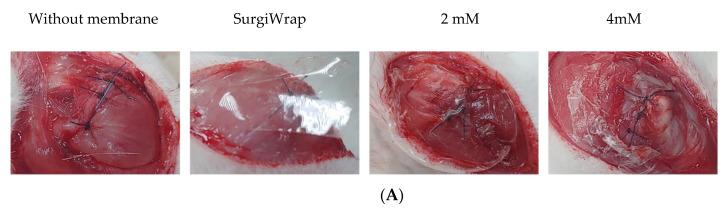
(**A**) Coverage of the GHF and SurgiWrap membranes after RC tear operation; the membranes impregnated with 2 and 4 mM farnesol fit smoothly on the tear sites. SurgiWrap was not malleable, preventing close attachment to the tear site; (**B**) SEM images of the SurgiWrap and GHF membranes before (upper lane, A: SurgiWrap, B: pure GHF membrane, C: 2 mM GHF membrane and D: 4 mM GHF membrane) and after (lower lane, E: SurgiWrap, F: pure GHF membrane, G: 2 mM GHF membrane and H: 4 mM GHF membrane) immersion in PBS solution for 24 h (surface section). RC: rotator cuff; GHF membranes: farnesol-impregnated hydrogel membranes containing hyaluronic acid or gellan gum; PBS: phosphate-buffered saline; SEM: scanning electron microscopy. Magnification: 500×.

**Figure 4 pharmaceutics-13-00944-f004:**
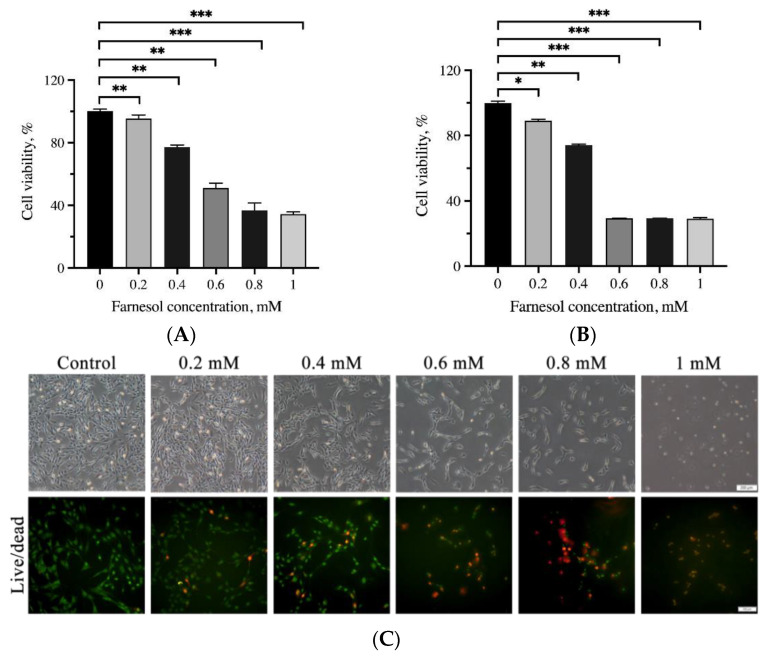
Cell viability determined using the MTT assay for (**A**) C2C12 myoblasts and (**B**) tenocytes after treatment with 0.2–1.0 mM farnesol over 24 h. Live/dead cell staining for (**C**) C2C12 myoblasts and (**D**) tenocytes. Green and red indicate live and dead cells, respectively. Significant differences are based on comparisons with the control group (without addition of farnesol), * *p* < 0.05, ** *p* < 0.01, and *** *p* < 0.001. Scale bar: 50 μm.

**Figure 5 pharmaceutics-13-00944-f005:**
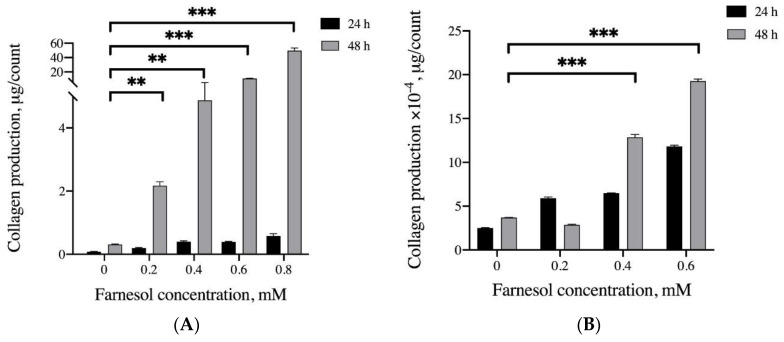
Effects of farnesol on collagen production in (**A**) C2C12 myoblasts and (**B**) tenocytes. Significant differences are based on comparisons with the control group after 48 h incubation (without addition of farnesol), ** *p* < 0.01 and *** *p* < 0.001. Effect of the anti-inflammatory response of farnesol induced by H_2_O_2_ exposure; (**C**) IL-6 and (**D**) TNF-α expression assayed in C2C12 myoblasts; (**E**) IL-6 and (**F**) NF-α expression assayed in tenocytes. * *p* < 0.05, ** *p* < 0.01, and *** *p* < 0.001. IL-6: interleukin 6; TNF-α: tumor necrosis factor α.

**Figure 6 pharmaceutics-13-00944-f006:**
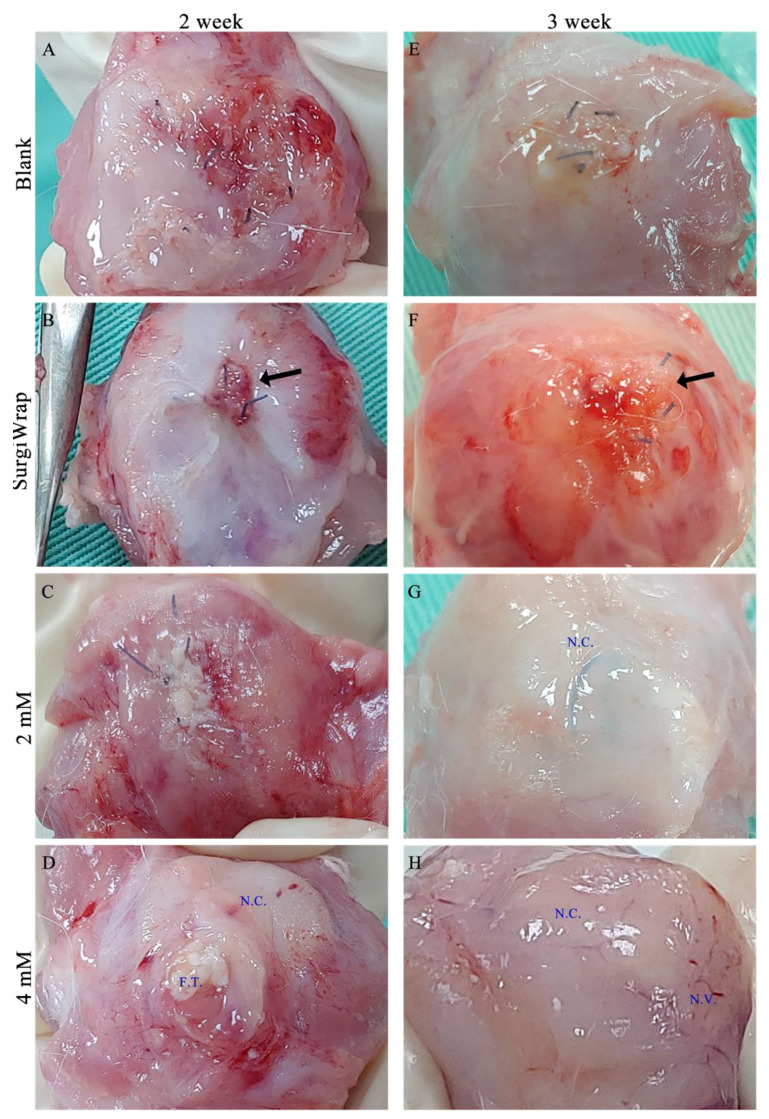
Gross observation of the RC tear after 2 weeks (**A**: without membrane, **B**: covered with SurgiWarp, **C**: covered with 2 mM farnesol-impregnated membrane and **D**: covered with 4 mM farnesol-impregnated membrane) and 3 weeks (**E**: without membrane, **F**: covered with SurgiWarp, **G**: covered with 2 mM farnesol-impregnated membrane and **H**: covered with 4 mM farnesol-impregnated membrane) of healing. Black arrows indicate sutured RC tears. RC: rotator cuff; F.T.: fibrous issue; N.C.: newly generated connective tissue; N.V.: newly generated vessel.

**Figure 7 pharmaceutics-13-00944-f007:**
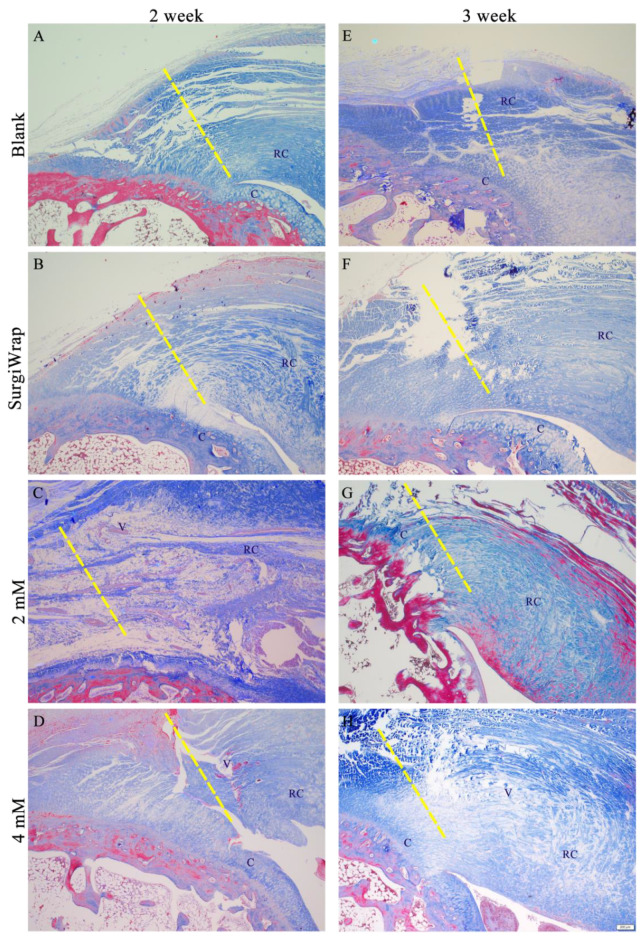
Masson’s trichrome staining of RC tear repair 2 weeks (**A**: without membrane, **B**: covered with SurgiWarp, **C**: covered with 2 mM farnesol-impregnated membrane and **D**: covered with 4 mM farnesol-impregnated membrane) and 3 weeks (**E**: without membrane, **F**: covered with SurgiWarp, **G**: covered with 2 mM farnesol-impregnated membrane and **H**: covered with 4 mM farnesol-impregnated membrane) postoperation after treatment with hydrogel membranes. RC: rotator cuff; V: vessel; C: cartilage. The dashed lines indicate RC tear sites. Scale bar: 200 μm.

**Figure 8 pharmaceutics-13-00944-f008:**
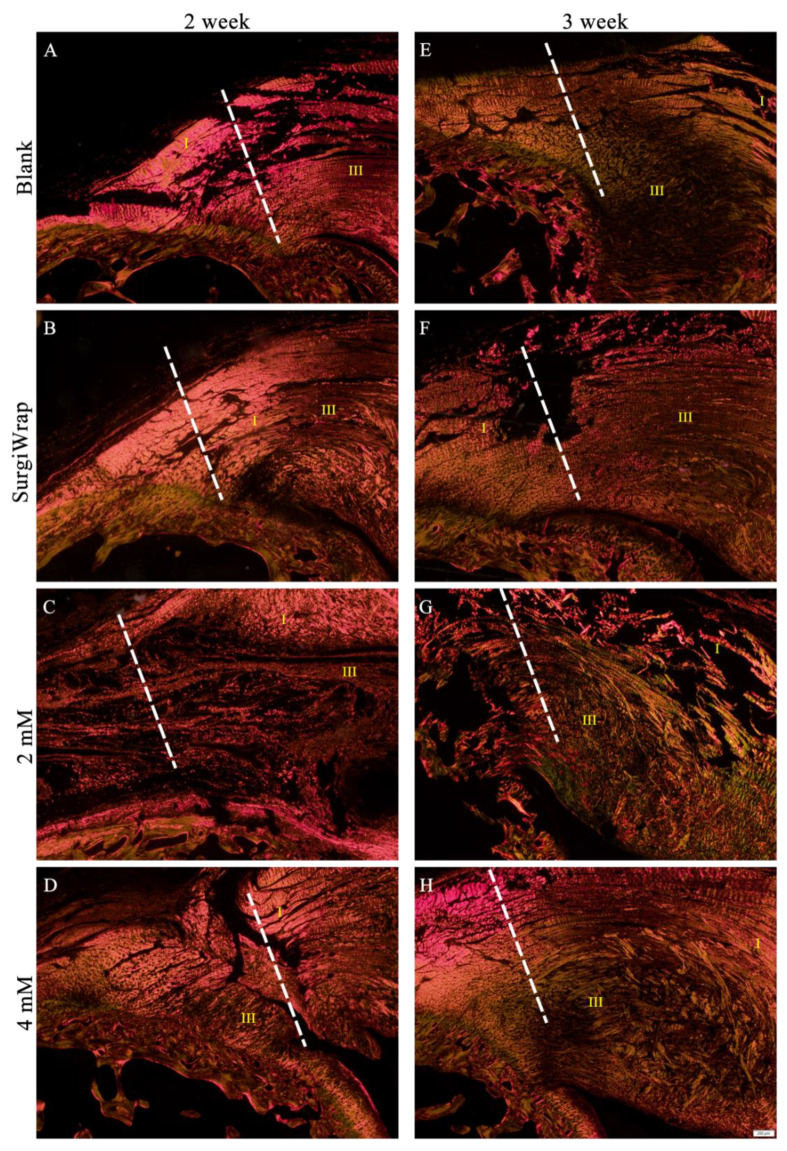
RC tears stained with picrosirius red to show type I collagen and type III collagen (bright red and green, respectively) 2 weeks (**A**: without membrane, **B**: covered with SurgiWarp, **C**: covered with 2 mM farnesol-impregnated membrane and **D**: covered with 4 mM farnesol-impregnated membrane) and 3 weeks (**E**: without membrane, **F**: covered with SurgiWarp, **G**: covered with 2 mM farnesol-impregnated membrane and **H**: covered with 4 mM farnesol-impregnated membrane) postoperation, after treatment with hydrogel membranes. RC: rotator cuff; I: type I collagen; III: type III collagen. The dashed lines indicate RC tear sites. Scale bar: 200 μm.

**Figure 9 pharmaceutics-13-00944-f009:**
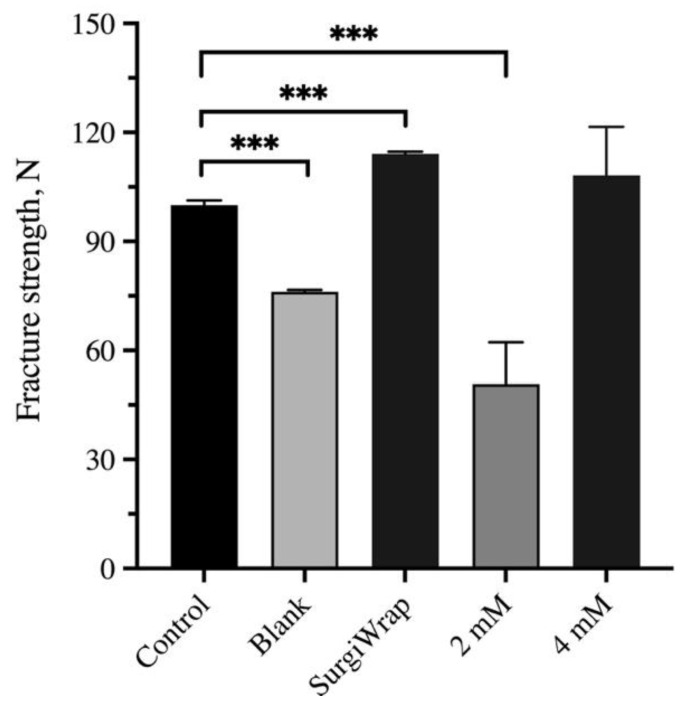
Mechanical strength was evaluated by determining the breaking force of the rotator cuff. Control: without transection, Blank: without membrane coverage. Significant differences are based on comparisons with the control group (normal RC without tear), *** *p* < 0.001.

**Table 1 pharmaceutics-13-00944-t001:** Modified histologic grading system for rotator cuff tear repair.

Score	Collagen Fiber Density	Collagen Fiber Orientation	Tendon–Bone Interface Interdigitation	Vascularity	Collagen I/III Ratio
1	None	None	<25%	None	None
2	Low	Disorganized	25–50%	Minimal	Minimal
3	Median	Moderately aligned	50–75%	Moderate	Moderate
4	High	Highly aligned	>75%	Abundant	Abundant

**Table 2 pharmaceutics-13-00944-t002:** Characteristics of the hydrogel and SurgiWrap membranes.

Group	Water Content, %	Swelling Ratio, %	Young’s Modulus, MPa	Max. Elongation, mm
0 mM	90.2 ± 0.4	923.3 ± 37.6	29.6 ± 4.8 *	1.7 ± 0.4
2 mM	89.0 ± 1.8	833.3 ± 16.0	50.7 ± 7.2 **	1.1 ± 0.03 ***
4 mM	94.8 ± 0.9	1860.8 ± 307.7	61.1 ± 7.9 **	1.0 ± 0.1 ***
SurgiWrap	36.8 ± 3.0	58.3 ± 7.2	40.7 ± 0.3	1.3 ± 1.5

Comparison with the SurgiWrap membrane. * *p* < 0.05, ** *p* < 0.01, and *** *p* < 0.001.

**Table 3 pharmaceutics-13-00944-t003:** Average scores obtained from hematoxylin and eosin staining and picrosirius red staining (modified from Table 1).

Time	Blank	SurgiWrap	2 mM	4 mM
2 week	6.67 ± 0.58	6.33 ± 2.51	8.33 ± 1.53 *	10.67 ± 1.15 **^,#^
3 week	11.67 ± 2.51	11.33 ± 1.51	14.33 ± 2.31 **	18.67 ± 0.58 ***^,##^

* Compared with the blank group, * *p* < 0.05, ** *p* < 0.01, *** *p* < 0.001. ^#^ Compared with 2 mM farnesol, ^#^
*p* < 0.05, ^##^
*p* < 0.01.

## Data Availability

The data presented in this study are available in the paper.

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
