# Peer review of "Enhancement of Rotator Cuff Healing with Farnesol-Impregnated Gellan Gum/Hyaluronic Acid Hydrogel Membranes in a Rabbit Model"

_pharmaceutics, 2021, doi:10.3390/pharmaceutics13070944_

Round 1

Reviewer 1 Report

Which CAS numbers are of the used chemicals in this work ?

Please put the CAS numbers in experimental part of the work.

Also, the English language and style are fine/minor spell check is required.

Reviewer 2 Report

Dear Authors,

Presented manuscript contains a lot of experiments, which were well described, but I found a few problems:

  1. Part 2.1. Fabrication of GHF Hydrogel Membranes - Please add word “kind”, because you did not prepare three pieces of membrane but three kind of membrane.
  2. Part „In Vitro Release Test” – You used only 5 ml of PBS as acceptor fluid and you took 1 ml at designated time points without replenishment the total volume!!! After 5 sampling you have nothing of PBS! Are you sure that you have sink condition when you use 5 ml of PBS as acceptor medium? Is it a mistake and it should be 50 ml?
  3. Parts „2.3. In Vitro Vell Viability”and “2.4. Quantification of Collagen Production” – In Results you showed the controls, but in Methods there is no information about controls. Please correct the word Vell to Cell.
  4. Results Figure 2D – Why did you show the release of API only from one formulation? You wrote “A shown in Figure 2D, the GHF membrane impregnated with 4 mM farnesol rapidly released farnesol in 24-h incubation under shaking at 40 rpm, after which it slowly released farnesol until day 9.” – Please transform this phrase e.g. as it was shown in Figure…” and add the information that after 24 h of diffusion about 60% of farnesol was released and after that …
  5. Figures 2 and 3 are to complex. Please consider dividing it into two.

Reviewer 3 Report

The authors present a very nice work related to the production of  a membrane  based on  gellan gum (GA), hyaluronic acid (HA) impregnated with farnesol to repair the rotator cuff (RC). The studied the physicochemical and biological  properties of the membranes and they were compared to the commercial SurgiWrap membrane. They also proved that membranes based on GA and HA impregnated with 4 mM farnesol resulted in superior RC repair.

* Specific comments:

-Line 85: the authors should provide the molecular weight of GA

- Line 125: Why the authors clain that it is an "in vitro degradation test"? No enzymes were used. I would rephrase the title to "In vitro stability test"

-Line 143: there is a typo. It is cell viability

-Line 146: why they authors measure the cell viability with only different concentrations of farnesol? Why they did not measure the cell viability over the membranes? It is a more real study.

-Lines 181-185: Could the authors clarify how many numbers of samples  of echa group was used? It is not clear.

-Lines 251-254, lines 264-267: Why the GHF membranes with 4 mM had the highest water content and swelling ratio and the same time they had a higher Young-modulus? It seems inconsistent with the explanation that they provide the authors: "farnesol inhibits the crosslinking reaction of the EDC/NHS reagent"....

-What is A, B, and C in figure 2A?

-The authors should improve the quality of the SEM images. They are very poor

- Line 323: Why the auhtors performed the  cell viability experiment at 24 hours? They should have performed at longer times, at lest 48-72 hours to see the effect of the farnesol concentration at longer times...

Round 2

Reviewer 3 Report

The authors addressed all my comments except the comment related to the quality of SEM images. I recommend to the authors to keep the SEM images of the first submission but they need to improve the Figure 2E.

Round 3

Reviewer 3 Report

Manuscript is ready for publication